# Association of Tumor Volumetry with Postoperative Outcomes for Cervical Paraganglioma

**DOI:** 10.3390/diagnostics13040744

**Published:** 2023-02-15

**Authors:** Carola Marie Hoffmann-Wieker, Artur Rebelo, Martin Moll, Ulrich Ronellenfitsch, Fabian Rengier, Philipp Erhart, Dittmar Böckler, Jörg Ukkat

**Affiliations:** 1Department of Vascular and Endovascular Surgery, University Hospital Heidelberg, 69120 Heidelberg, Germany; 2Department of Visceral, Vascular and Endocrine Surgery, University Hospital Halle (Saale), 06097 Halle (Saale), Germany; 3Department of Diagnostic and Interventional Radiology, University Hospital Heidelberg, 69120 Heidelberg, Germany

**Keywords:** cervical paraganglioma, carotid body tumor, preoperative imaging, volumetry, cranial nerve injury

## Abstract

**Objectives:** To analyze the association of tumor volume with outcome after surgery for cervical paraganglioma. **Materials and Methods:** This retrospective study included consecutive patients undergoing surgery for cervical paraganglioma from 2009–2020. Outcomes were 30-day morbidity, mortality, cranial nerve injury, and stroke. Preoperative CT/MRI was used for tumor volumetry. An association between the volume and the outcomes was explored in univariate and multivariable analyses. A receiver operating characteristic (ROC) curve was plotted, and the area under the curve (AUC) was calculated. The study was conducted and reported according to the STROBE statement. **Results:** Volumetry was successful in 37/47 (78.8%) of included patients. A 30-day morbidity occurred in 13/47 (27.6%) patients with no mortality. Fifteen cranial nerve lesions occurred in eleven patients. The mean tumor volume was 6.92 cm^3^ in patients without and 15.89 cm^3^ in patients with complications (*p* = 0.035) and 7.64 cm^3^ in patients without and 16.28 cm^3^ in patients with cranial nerve injury (*p* = 0.05). Neither the volume nor Shamblin grade was significantly associated with complications on multivariable analysis. The AUC was 0.691, indicating a poor to fair performance of volumetry in predicting postoperative complications. **Conclusions:** Surgery for cervical paraganglioma bears a relevant morbidity with a particular risk of cranial nerve lesions. Tumor volume is associated with morbidity, and MRI/CT volumetry can be used for risk stratification.

## 1. Introduction

Paragangliomas are vascularized neoplasms that derive from the neural crest and occur sporadically or due to a hereditary predisposition [1,2,3,4]. They commonly occur in the cervical region [5]. Carotid body tumors (CBTs) are paragangliomas located in the carotid bifurcation [6]. Their incidence is estimated at 1/100,000 persons per year. Most of them have a benign biological behavior, but 5–16% of these tumors show malignant transformation and metastasis. Lymphatic as well as distant metastases can occur [7]. The risk of metastasis, multilocular appearance and recurrences represent challenging aspects in the care of these patients [8]. A definite differentiation between benign and malignant lesions can only be made by a histopathological assessment of the specimen, and resection is therefore generally recommended [9].

Preoperative imaging of cervical paraganglioma allows to confirm the diagnosis, identify multifocal disease, and determine the extent of the tumor. Magnetic resonance imaging (MRI) including MR angiography is considered the non-invasive imaging modality of choice, but computed tomography (CT) angiography is an appropriate alternative [10,11,12]. 

Because of the close proximity to vascular and nerval structures, surgical resection of cervical paraganglioma can be challenging. There is a relevant risk of resection-related neurological complications, such as cranial nerve injury in 27–53.8% of patients [13,14]. Based on preoperative imaging, cervical paraganglioma is commonly described using the Shamblin classification. This system stratifies tumors according to the extent of their anatomic contact with the carotid vessels in three groups: I = minimal contact to III = full encasement [15]. This classification shows a good correlation with postoperative morbidity and cranial nerve injury [9]. However, it remains unclear if other features, which can be assessed on preoperative imaging, such as tumor volume and tumor location, also correlate with postoperative morbidity.

The aim of this study was to explore a possible association between tumor volume and tumor location determined on preoperative imaging and postoperative outcomes for the resection of cervical paraganglioma.

## 2. Materials and Methods

### 2.1. Study Design

This retrospective study comprised consecutive patients undergoing open surgical therapy for cervical paraganglioma in two vascular centers between January 2009 and January 2020. Demographic and clinical data as well as follow-up data were collected retrospectively from patient charts, hospital information systems, and Picture Archiving and Communication Systems (PACS). Patients were followed up routinely in hospital and in an outpatient clinic 30 days postoperatively. 

The study was conducted according to the guidelines of the Declaration of Helsinki and approved by the competent ethics committee (Medical Faculty of the University of Heidelberg, Germany, reference number: S-026/2021; approval date: 26 February 2021). The study is reported according to the guidelines of the STROBE statement (Appendix A) [16].

### 2.2. Endpoints and Definitions

Endpoints were 30-day morbidity, 30-day incidence of cranial nerve injury and Horner syndrome, 30-day stroke incidence and 30-day mortality. Morbidity was adjudicated by the investigators based on the exams and treatment noted. All cranial nerve lesions had to be determined by an ENT specialist or neurologist. Nerve lesions were considered temporary if the associated symptoms had subsided at 30-day follow-up and permanent if the symptoms prevailed. Stroke was defined as any new-onset neurological deficit lasting more than 24 h, diagnosed by a neurologist. A neurological impairment lasting less than 24 h was considered a transient ischemic attack (TIA).

### 2.3. Image Analysis

Preoperative contrast-enhanced CT or contrast-enhanced MRI of the neck region, depending on availability, were used as preoperative imaging. In the case of bilateral tumors, each side was analyzed individually. A three-dimensional tumor segmentation for tumor volume calculation was performed using three-dimensional image processing software (mint Lesion^TM^ software platform, v3.4.5; Mint Medical) by manual delineation of the tumor margins (Figure 1). 

The carotid arteries were included into the segmentation if the encasement was ≥180°. The axial slice with the largest tumor area, as defined by the three-dimensional tumor segmentation, was identified, and the long and short axis diameters were recorded at this slice. The same slice was used to manually measure the distance between the internal and external carotid arteries. Furthermore, the encasements of the internal and the external carotid arteries were evaluated and visually graded as 0–89°, 90–179°, 180–269°, 270–359° or 360°. Finally, the level of the carotid bifurcation was assessed in relation to the spine. Using sagittal reconstructions, a line was drawn perpendicularly to the spine through the upper border of the carotid bifurcation to identify the level of the carotid bifurcation at the upper/mid/lower third of the respective vertebra or at the intervertebral disc. 

Volumetry was performed by one board-certified fellow radiologist in consensus with a board-certified attending radiologist. Both radiologists were blinded to the outcomes of patients.

### 2.4. Surgical Technique 

All procedures were conducted in general anesthesia and were performed by a board-certified vascular surgeon alone or in an interdisciplinary team with an ENT specialist. In most cases, tumors were dissected in a periadventitial plane by using a bipolar knife to avoid bleeding. If an unplanned vascular reconstruction necessitating clamping of the internal carotid artery was performed, transcranial oxygen saturation measurement (Invos^®^ Cerebral Oximeter) was used. In such cases, completion angiography was performed to rule out stenosis, dissection, or thrombosis of the vascular reconstruction. Selected patients underwent preoperative angiography to attempt the embolization of tumor-feeding vessels.

### 2.5. Statistical Analysis

Descriptive data were given as a mean ± standard deviation and as a median and interquartile range in the case of non-parametric data. Continuous data were compared using the Mann–Whitney-U test. Proportions were compared using the Fisher-exact test (if there were fewer than five observations per category) or chi-square test. Patients with missing information for single variables were not included in the respective analyses. A multivariable logistic regression analysis was performed with postoperative complications as the dependent variable and with age, sex, and the two significant predictors on univariate analysis, quartile of tumor volume and Shamblin grade, as independent variables. Only patients with information for all used variables were included in the analysis. Goodness-of-fit was assessed with the Hosmer–Lemeshow test. All *p* values (significance level *p* < 0.05) and 95% confidence intervals (CIs) were two-sided. A receiver operating characteristic (ROC) curve was plotted and the area under the curve (AUC) calculated to assess the diagnostic performance of volumetry as a predictor of postoperative complications [17].

## 3. Results

### 3.1. Patient Characteristics and Procedural Results

The study included 47 patients (mean age 49 years, range 17–77 years, 63.8% female). All patients had carotid body paraganglioma. Tumor characteristics and demographics of the patients are shown in Table 1. Three patients (6.4%) showed preoperative symptoms, such as local pain, dysphagia, and hoarseness. The Shamblin classification was ascertained in 97.9% of patients (*n =* 46): most patients (*n =* 19, 40.4%) had a type I tumor followed by type III (*n =* 14, 29.8%) and type II (*n =* 13, 27.6%). 

Five patients (10.6%) underwent successful preoperative embolization. In one patient (2.1%), preoperative angiography with unsuccessful embolization was performed; the patient suffered from an intraprocedural stroke. Six (12.7%) patients had a prior cervical surgical intervention. In five (10.6%) of these patients, this was a prior paraganglioma resection. In one patient, only cervical lymph nodes had been removed before without a resection of the paraganglioma.

In all patients, a complete tumor resection was technically successful. Two tumors showed criteria of malignancy on histopathology (4.3%). In a seventeen-year-old patient, systemic metastasis was observed; this patient underwent postoperative chemoradiotherapy because of pulmonary and bone metastases. Two patients had bilateral tumors (4.3%), and in one patient with a family history of paragangliomas, an SDHD-gene mutation was found.

In 17% (*n =* 8) of patients, surgery was performed in an interdisciplinary team with ENT specialists. The mean procedural time was 132 min. In four patients (8.5%), a vascular reconstruction of the internal carotid artery was necessary. Three patients (6.4%) received an alloplastic carotid interposition graft and one patient an autologous interposition graft. In one patient with preoperative embolization and endovascular occlusion of the internal carotid artery, a resection of the internal carotid artery with reconstruction of the external carotid artery was performed. Operative characteristics are summarized in Table 2.

### 3.2. Results of Study Endpoints

Thirty-day mortality was 0%. The overall perioperative complication rate was 27.6% (*n =* 13). Fifteen cranial nerve lesions in eleven patients (23.4%) were observed: four hypoglossal nerve, seven vagal nerve, three glossopharyngeal nerve and one facial nerve lesion. Three patients (6.4%) suffered from postoperative Horner syndrome. In eight patients, the cranial nerve lesion was permanent at 30-day follow-up. Three patients with a perioperative cranial nerve lesion were lost to follow-up. 

Thirty-day stroke incidence was 4.2% (*n =* 2). One patient had an infiltrative tumor requiring an autologous interposition graft; in addition, internal carotid artery stenting at the skull base was necessary. Postoperatively, the patient developed hemiparesis and aphasia. The second patient had an uneventful intraoperative course and developed immediate postoperative sensory aphasia. Both patients still showed neurological symptoms at 30-day follow-up.

### 3.3. Association with Imaging Findings

Duplex imaging was performed in all patients. Among the 47 included patients, preoperative MRI (63.8%, *n =* 30) and/or CT (55.3%, *n =* 26) was conducted. The volume measurement was successful in 78.8% (*n =* 37/47): eight CT and twenty-nine MRI. The mean horizontal tumor extension was 2.9 cm (range 1.1–5.5 cm), and the mean vertical extension was 2.2 cm (range 0.8–3.9 cm). The degree of encasement of the carotid vessels (Shamblin classification) showed no significant correlation with cranial nerve injury (*p* = 0.44) but did so with overall postoperative morbidity (cranial nerve injury, Horner syndrome, and stroke) (*p* = 0.037). 

There was a significant difference in tumor volume, as ascertained by volumetry, between patients with and without overall postoperative complications. Patients without complications had a mean tumor volume of 6.92 cm^3^, while patients with complications had one of 15.89 cm^3^ (*p* = 0.035). The mean tumor volume was 7.64 cm^3^ in patients with no cranial nerve injury and 16.28 cm³ in patients with cranial nerve injury (*p* = 0.05). Most tumors projected to the fourth cervical vertebra (*n =* 18; 38.3%) and were located at the lower third of the vertebra. There was no significant correlation between the tumor location in relation to the vertebra and the occurrence of cranial nerve lesions (*p* = 0.42). The degree of encasement of the carotid vessels showed no significant correlation with cranial nerve injury (*p* = 0.44). The results are summarized in Table 3. 

The results of the multivariable logistic analysis, which was based on 35 cases with complete information for all variables, are displayed in Table 4. Neither the tumor volume nor the Shamblin grade showed a significant association with postoperative complications on multivariable analysis.

The receiver operating characteristic (ROC) curve yielded an area under the curve (AUC) of 0.691, which indicates a poor to fair performance of volumetry as a predictor of postoperative complications (Figure 2).

## 4. Discussion

The present study aimed to evaluate the utility of preoperative-imaging-based volumetry in assessing the risk of perioperative morbidity and cranial nerve injury in patients undergoing open surgery for cervical paraganglioma. The results show that the preoperative tumor volume is associated with surgical morbidity and that MRI/CT volumetry using a dedicated radiological software can be used for risk stratification as an adjunct to the long-established Shamblin classification.

In 2017, Kim et al. analyzed the relationship of the Shamblin grade, tumor distance to the base of the skull and tumor volume with complications from cervical paraganglioma resection, including bleeding and cranial nerve injury. A total of 332 patients with 356 resections were included. Similar to the results of the present study, the most commonly injured cranial nerves were the hypoglossal and vagal nerves (11% and 10%). Both the Shamblin grade and tumor distance to base of the skull were associated with blood loss and cranial nerve injuries, whereas the tumor volume was associated only with blood loss but not with cranial nerve injuries [18].

In 2022, Ivanjko et al. aimed to confirm the findings reported by Kim et al. The authors analyzed the effect of the distance to the base of the skull and tumor-size characteristics on cranial nerve injuries in carotid body tumor resections. A total of 48 CBTs were included. The distance to the base of the skull, craniocaudal tumor diameter, and tumor volume were statistically significantly associated with cranial nerve lesions on univariate analysis, while the distance to the base of the skull was the only parameter that retained significance on multivariable analysis. While in the study by Ivanjko et al. cranial nerve lesions occurred in 37.5% of patients, in our study population this was the case for 23.4% of patients. Contrary to our study, in which the vagal nerve was the most affected cranial nerve, in Ivanjko et al.’s study population this was the case for the hypoglossal nerve. The main methodological difference between the two studies is that our study assessed not only cranial nerve lesions but all postoperative complications, including clinically important events such as stroke or hemorrhage. Moreover, volumetry in Ivanjko’s study was calculated using a rigid formula, whereas in our study it was determined with a dedicated radiological software [19].

In 2012, Power et al. analyzed 132 patients with 144 cervical paraganglioma resections. The authors determined that the most common postoperative complication was temporary cranial nerve injury and that it was significantly associated with the tumor volume, which was however calculated using three axes and no dedicated algorithm. Thirty-three percent of patients suffered from cranial nerve injury, and the majority of patients (58%) had Shamblin type III tumors. Preoperative embolization, the operating time, and a greater blood loss were also associated with temporary cranial nerve injury [20].

In line with these previous studies, we can confirm that the resection of cervical paraganglioma bears a relevant risk of morbidity and, in particular, of cranial nerve injury and stroke. Resection is recommended for all tumors of the carotid bifurcation region in light of their potentially malignant behavior. However, given that most cervical paragangliomas are asymptomatic and of benign histology, a thorough risk-benefit assessment needs to be performed, and procedural risks need to be well discussed with patients prior to surgery. The findings of our study show that, in addition to the established Shamblin classification, which was also significantly associated with the study outcome perioperative complications, tumor volume measured with a dedicated software on the basis of preoperative cross-sectional imaging can be used to predict the risk of perioperative complications, with larger tumors bearing a higher risk of complications. This information can be used in shared decision-making with patients [21]. Moreover, surgeons can anticipate the procedural risks based on tumor volumetry and take particular precautions during the operation. In a multivariable analysis, we attempted to determine which of the two modalities, volumetry or the Shamblin classification, is a more suitable predictor for postoperative complications. However, neither of the two showed significance on the analyses, which is probably due to collinearity and the rather small sample size included in the multivariable analyses. The receiver operating characteristic (ROC) curve showed a poor to fair performance of the tumor volume as a predictor of postoperative complications, which shows that this measure is not a perfectly discriminating test, but rather an indicative predictor.

This study has some methodological limitations. It has a retrospective design, and volumetry was conducted retrospectively using available imaging material, which was not acquired with the explicit aim of volumetry. Therefore, volumetry was not in all patients technically possible. However, the imaging assessment followed a defined protocol, and assessors were blinded towards patients’ outcomes. Data on outcomes were extracted from prospectively kept institutional databases but might have still been incomplete for some patients. Nevertheless, a systematic bias regarding the completeness of data seems unlikely. Cranial nerve lesions were ascertained by ENT specialists not involved in the surgical treatment of patients, thus reducing the risk of observer bias. The sample size and thus statistical power of the study was rather small. It is a strength of the study that all consecutive patients undergoing resection of cervical paraganglioma in the two participating institutions were included and analyzed, thus minimizing a possible selection bias and increasing the external validity of the findings. 

## 5. Conclusions

This study shows an association between the tumor volume of cervical paraganglioma and postoperative morbidity on univariate analysis, which loses significance when adjusting for other covariables such as the Shamblin classification. Tumor volume can be used as additional information in a risk-benefit analysis and discussions with patients prior to cervical paraganglioma resection. Volumetry should be considered to become part of routine preoperative diagnostics prior to cervical paraganglioma resection.

## Figures and Tables

**Figure 1 diagnostics-13-00744-f001:**
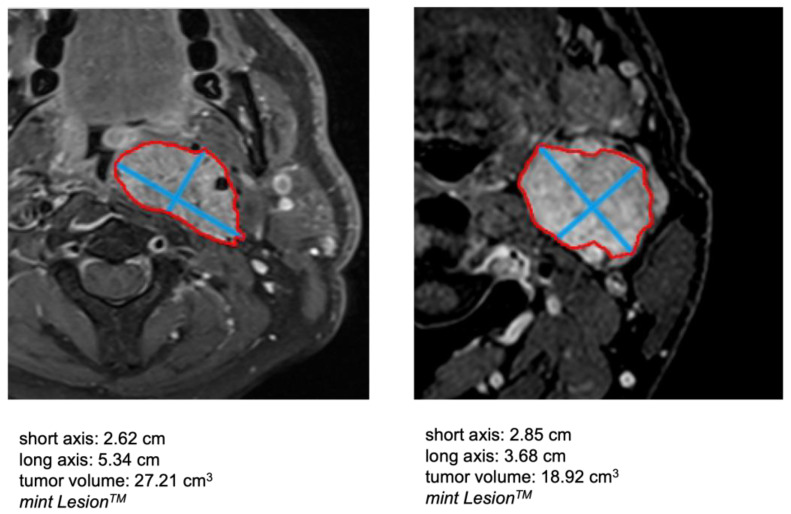
Three-dimensional tumor segmentation for tumor volume calculation (mint Lesion^TM^).

**Figure 2 diagnostics-13-00744-f002:**
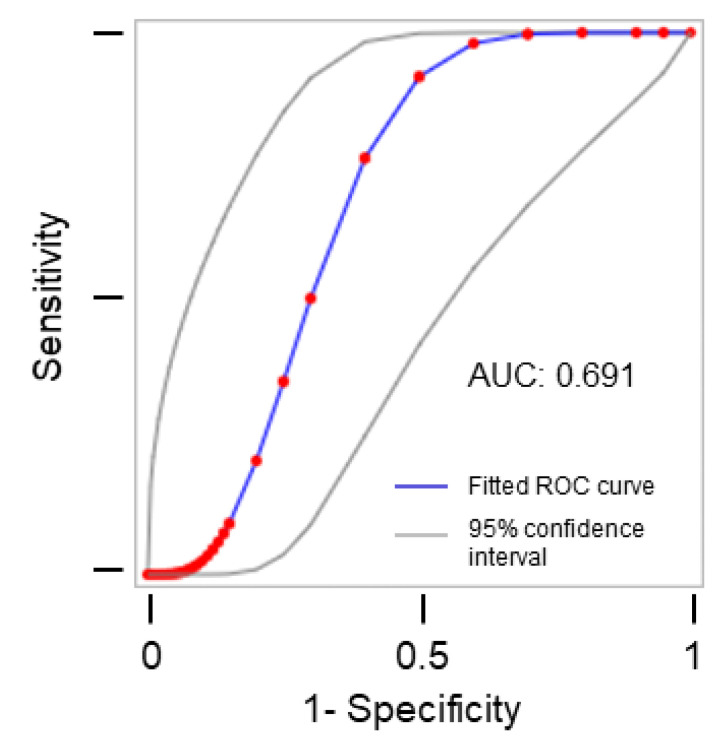
Receiver operating characteristic (ROC) curve for the diagnostic performance of volumetry as a predictor of overall postoperative complications. AUC: area under the curve.

**Table 1 diagnostics-13-00744-t001:** Tumor characteristics and demographics. MRI: magnetic resonance imaging; CT: computed tomography.

	*n =* 47	Percentage (%)
age (median, range)	49 (17–77)	
gender		
female	30	63.8
male	17	36.2
Shamblin classification		
type I	19	40.4
type II	13	27.6
type III	14	29.8
not assessable	1	2.1
preoperative imaging		
Duplex scan	32	68.1
MRI scan	30	63.8
CT scan	26	55.3
angiography	1	2.1
time until diagnosis (mean)	20 months	
preoperative symptoms	3	3.4
preoperative embolization attempt	6	12.8

**Table 2 diagnostics-13-00744-t002:** Operative characteristics. ICA: internal carotid artery; CAS: carotid artery stenting; ENT: ear, nose, and throat specialist.

	*n =* 47	Percentage (%)
redo operation	*n =* 6	12.7
operating time (mean)	132 min	
general anesthesia	*n =* 47	100
interposition graft ICA	*n =* 4	8.5
autologous	*n =* 1	2.1
allogenous	*n =* 3	6.4
ICA resection	*n =* 1	2.1
CAS	*n =* 1	2.1
interdisciplinary operation with ENT specialist	*n =* 8	17.0

**Table 3 diagnostics-13-00744-t003:** Comparison of volumetry and imaging characteristics between patients with and without postoperative complications. ICA: internal carotid artery.

	Perioperative Complications (*n =* 13)	No Perioperative Complications (*n =* 34)	*p*-Value
tumor volume (cm^3^/median)	15.89	6.92	0.035
Shamblin classification			
type I	2	17	
type II	3	10	
type III	8	6	0.016
tumor localization in projection to cervical vertebrae			
cervical vertebrae 2/3	6	9	
cervical vertebrae 4/5	6	16	0.416
tumor encasement of the carotid arteries			
0–89°	0	0	
90–179°	2	9	
180–169°	3	8	
270–359°	2	3	
360°	5	5	0.442
ICA interposition graft			
yes	2	2	
no	11	32	0.304
preoperative embolization			
yes	3	3	
no	9	31	0.173

**Table 4 diagnostics-13-00744-t004:** Results of logistic regression analysis with postoperative complications as dependent variable. Hosmer–Lemeshow test: *p* = 0.84.

Variable	Category	OR	95% CI
Age (continuous)		0.97	0.91–1.03
Sex	female	reference	
	male	1.14	0.17–7.54
Shamblin classification	1	reference	
	2	2.54	0.18–34.89
	3	12.10	0.94–156.35
Tumor volume	per quartile	1.63	0.65–4.09

## Data Availability

Data available on request due to privacy restrictions.

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
