# Peer review of "Association of Tumor Volumetry with Postoperative Outcomes for Cervical Paraganglioma"

_diagnostics, 2023, doi:10.3390/diagnostics13040744_

Round 1

Reviewer 1 Report

This manuscript is an original study, correctly conceived and carried out. The main focus is establishing a relationship between the tumour volume and location, and several postoperative outcomes.

Minor recommendations:

51-53: Using such a long quote is not desirable. I encourage the authors to find a way to convey this message.

Introduction:  Considering that the study starts from the imaging evaluation of tumours, it would be useful to have a paragraph in the introduction that addresses this aspect emphasizing the state of knowledge regarding the best imaging method, and the correlation between preoperative imaging and the postoperative outcome of patients.

 I recommend improving the discussion section: a more detailed integration of the results in the already existing literature.

 Thank you!

Author Response

Reviewer #I

- 51-53: Using such a long quote is not desirable. I encourage the authors to find a way to convey this message.

Response: Redline Manuscript: Lines 82-85; The quote was shortened and changed.

"Based on preoperative imaging, cervical paraganglioma is commonly described using the Shamblin classification. This system stratifies tumors according to the extent of their anatomic contact with the carotid vessels in three groups: I = minimal contact to III = full encasement [15].”

- Introduction:  Considering that the study starts from the imaging evaluation of tumours, it would be useful to have a paragraph in the introduction that addresses this aspect emphasizing the state of knowledge regarding the best imaging method, and the correlation between preoperative imaging and the postoperative outcome of patients. 

Response: Redline Manuscript: Lines 74-78; According to your suggestion, we have added a pertianing paragraph to the introduction.

- I recommend improving the discussion section: a more detailed integration of the results in the already existing literature.

Response: Redline Manuscript: Lines 251-268; The discussion has been extended accordingly reflecting our results in relationship to the existing evidence on the topic.

- If I understand well, one patient has been evaluated by angiography. This kind of evaluation is not reported in the methods section. Moreover, it seems complicated to perform volumetry by angiography. This patient has to be exclude from the analysis.

Response: Redline Manuscript: Lines 150-151; this is a misunderstanding; this specific patient underwent angiography for planned embolization, which was however not successful. Volumetry for this patient has been done using cross-sectional imaging, and thus the patient has been included in the analyses. We have added the following in the “Surgical Technique section”:

“Selected patients underwent preoperative angiography to attempt embolization of tumor-feeding vessels.”

- Is the genetic status of all patients evaluated? The authors have to report this kind of data.

Response: No, given the long inclusion period dating back to 2009, the genetic status was unfortunately not evaluated for the majority of patients and could thus not be reported

- The correlations will be more accurate with a univariate and multivariate analyses.

Response: Following your suggestion, we have added a multivariable analysis to the existing univariate analyses, as described in the revised methods and results section. We now also discuss the results of the multivariable analysis in the discussion section.

- In the discussion section: the authors have to discuss why they do not have correlation with Shamblin classification.

Response: Redline Manuscript: Lines 284-289;

This seems to be a misunderstanding. We did find a correlation of postoperative complications with the Shamblin classification, and this had already been addressed in the discussion section. We have now revised the wording to emphasize this finding:

"The findings of our study show that, in addition to the established Shamblin classification, which also was significantly associated with the study outcome perioperative complications, tumor volume measured on the basis of preoperative cross-sectional imaging can be used to predict the risk of perioperative complications with larger tumors bearing a higher risk of complications.”

- Page 5 line 196: this sentence has to be more precise.  

Response: Redline Manuscript: Lines 278-279; the sentence was specified:

“Resection is recommended for all tumors of the carotid bifurcation region given their potentially malignant behavior.”

Reviewer 2 Report

Thank you very much for the opportunity of reviewing this study. As one would expect from this group, investigation was carefully conducted. However, this study has some problems.

I wrote the message below to ask you.

I look forward to your reply to my thoughts.

#1 The authors mentioned tumor volume is associated with surgical morbidity and that radiological findings can be used for risk stratification. If risk factors are evaluated, the author should perform the cutoff points using ROC curve and logistic analysis.

#2 Since it is an observational study, I recommend to use the STROBE statement to improve the reporting.

#3 There were no result of duplex sonography in abstract and manuscript.

#4 The author should show inter-rater reliability and intraclass correlation coefficients about tumor volume.

Author Response

Reviewer #II

Thank you very much for the opportunity of reviewing this study. As one would expect from this group, investigation was carefully conducted. However, this study has some problems.

I wrote the message below to ask you.

I look forward to your reply to my thoughts.

#1 The authors mentioned tumor volume is associated with surgical morbidity and that radiological findings can be used for risk stratification. If risk factors are evaluated, the author should perform the cutoff points using ROC curve and logistic analysis.

Response: Redline Manuscript: Abstract Lines 18-19 and 24-26; Results Lines 224-230; Figure 2; Discussion Lines 294-302; Following your suggestion, a receiver operating characteristic (ROC) curve was plotted and the area under the curve (AUC) calculated to assess the diagnostic performance of volumetry as a predictor of postoperative complications. Moreover, a logistic regression analysis has been added.

#2 Since it is an observational study, I recommend to use the STROBE statement to improve the reporting.

Response: Redline Manuscript: Line 19, supplemental figure 1

Thank you for this suggestion. We have amended a STROBE checklist as supplementary material to the manuscript.

#3 There were no result of duplex sonography in abstract and manuscript.

Response:

All patients received preoperative ultrasound examinations, but volumetry was exclusively performed using cross-sectional imaging. Therefore, no findings from duplex sonography were included in the analyses.

#4 The author should show inter-rater reliability and intraclass correlation coefficients about tumor volume.

Response:

All volumetry analyses were performed by one board-certified fellow radiologist in consensus with one board-certified attending radiologist, who were both blinded to the outcomes of patients. However, there were no independent volumetry assessments by two or more radiologists. Therefore, no inter-rater reliability and intraclass correlation coefficients could be calculated.

Reviewer 3 Report

Hoffmann-Wieker et al. report a small retrospective cohort of cervical paragangliomas (PGL). The authors evaluated the correlation between the PGL volumetry and the surgical outcome. It is an interesting article on an important problem in the cervical PGL management’s.

- In the introduction section, line 38-41: the sentences are not clear or not precise enough.

- In the methods section: how PGL volumetries were evaluated? Single or multiple radiologist(s)?

- In the results section: is there a correlation between vascular surgery and study endpoints? Is there a correlation between metastasis and study endpoints? Is there a correlation between embolization and study endpoints?

- The different localizations of PGL have to be describe in the article (i.e., carotid body PGL, vagal PGL, tympanic PGL…) and correlation between localization and study endpoints has to be report.

- If I understand well, one patient has been evaluated by angiography. This kind of evaluation is not reported in the methods section. Moreover it seems complicated to perform volumetry by angiography. This patient has to be exclude from the analysis.

- Is the genetic status of all patients evaluated? The authors have to report this kind of data.

- The correlations will be more accurate with a univariate and multivariate analyses.

- In the discussion section: the authors have to discuss why they do not have correlation with Shamblin classification.

- Page 5 line 196: this sentence has to be more precise.  

Author Response

Reviewer #III

- In the introduction section, line 38-41: the sentences are not clear or not precise enough.

Response: Redline Manuscript: Lines 63-66; The sentences were changed.

"Carotid body tumors (CBT) are paragangliomas located in the carotid bifurcation [6]. Their incidence is estimated as 1/100,000 persons per year. Most of them have a benign biological behavior, but 5-16% of these tumors show malignant transformation and metastasis."

- In the methods section: how PGL volumetries were evaluated? Single or multiple radiologist(s)? 

Response: Redline Manuscript: Lines 138-141

An explanation how volumetries were performed has been added to the methods section.

- In the results section: is there a correlation between vascular surgery and study endpoints? Is there a correlation between metastasis and study endpoints? Is there a correlation between embolization and study endpoints?

Response: table 3

We are not fully clear what a correlation between “vascular surgery” and study endpoints would exactly mean. All patients included in the study underwent vascular surgery. We have now assessed a correlation between ACI interposition graft, i.e. more complex vascular surgery, and complications, which is not statistically significant (table 3). Likewise, we have now assessed a correlation between preoperative embolization and complications, which is also statistically not significant (table 3). Metastatic disease was present in only one patient, who had no complications, but obviously no correlation between metastasis and complications could be assessed.

- The different localizations of PGL have to be describe in the article (i.e., carotid body PGL, vagal PGL, tympanic PGL…) and correlation between localization and study endpoints has to be report.

Response: Redline Manuscript: Line 169; All included patients had carotid body PGL. This has now been clarified in the results section.

- If I understand well, one patient has been evaluated by angiography. This kind of evaluation is not reported in the methods section. Moreover, it seems complicated to perform volumetry by angiography. This patient has to be exclude from the analysis.

Response: Redline Manuscript: Lines 150-151; this is a misunderstanding; this specific patient underwent angiography for planned embolization, which was however not successful. Volumetry for this patient has been done using cross-sectional imaging, and thus the patient has been included in the analyses. We have added the following in the “Surgical Technique section”:

“Selected patients underwent preoperative angiography to attempt embolization of tumor-feeding vessels.”

- Is the genetic status of all patients evaluated? The authors have to report this kind of data.

Response: No, given the long inclusion period dating back to 2009, the genetic status was unfortunately not evaluated for the majority of patients and could thus not be reported

- The correlations will be more accurate with a univariate and multivariate analyses.

Response: Following your suggestion, we have added a multivariable analysis to the existing univariate analyses, as described in the revised methods and results section. We now also discuss the results of the multivariable analysis in the discussion section.

- In the discussion section: the authors have to discuss why they do not have correlation with Shamblin classification.

Response: Redline Manuscript: Lines 284-289;

This seems to be a misunderstanding. We did find a correlation of postoperative complications with the Shamblin classification, and this had already been addressed in the discussion section. We have now revised the wording to emphasize this finding:

"The findings of our study show that, in addition to the established Shamblin classification, which also was significantly associated with the study outcome perioperative complications, tumor volume measured on the basis of preoperative cross-sectional imaging can be used to predict the risk of perioperative complications with larger tumors bearing a higher risk of complications.”

- Page 5 line 196: this sentence has to be more precise.  

Response: Redline Manuscript: Lines 278-279; the sentence was specified:

“Resection is recommended for all tumors of the carotid bifurcation region given their potentially malignant behavior.”

Round 2

Reviewer 2 Report

I believe that the authors have answered the reviewers' questions appropriately. This manuscript will be worthy of publication.

Author Response

Thank you for accepting the manuscript.

Reviewer 3 Report

I would like to thank the authors for their answers.

I do not find the different tables in the revised manuscript.

Multivariate analysis do not show a strong association between volumetry and postperative complication. However the authors write in the conclusion that their study shows a significant association between tumor volume and postoperative morbidity. The conclusion has to be change according to the multivariate analysis.

Author Response

Reviewer #III

I do not find the different tables in the revised manuscript.

Response: we have now added the tables to the redline manuscript.

Multivariate analysis do not show a strong association between volumetry and postperative complication. However the authors write in the conclusion that their study shows a significant association between tumor volume and postoperative morbidity. The conclusion has to be change according to the multivariate analysis.

Response: Redline Manuscript: The conclusion was changed; Lines 295-301